# Evolution of the Cross-Sectional Area of the Osseous Lumbar Spinal Canal across Decades: A CT Study with Reference Ranges in a Swiss Population

**DOI:** 10.3390/diagnostics13040734

**Published:** 2023-02-15

**Authors:** Benoit Maeder, Fabio Becce, Sam Kehtari, Arnaud Monier, Etienne Chaboudez, Dominique A. Rothenfluh, Constantin Schizas, Steven D. Hajdu

**Affiliations:** 1Spine Center, Lausanne University Hospital and University of Lausanne, 1011 Lausanne, Switzerland; 2Department of Diagnostic and Interventional Radiology, Lausanne University Hospital and University of Lausanne, 1011 Lausanne, Switzerland; 3Department of Orthopedics and Traumatology, Lausanne University Hospital and University of Lausanne, 1011 Lausanne, Switzerland; 4Faculty of Biology and Medicine, University of Lausanne, 1011 Lausanne, Switzerland; 5Neuro-Orthopaedic Spine Unit, Clinic Cecil, 1003 Lausanne, Switzerland

**Keywords:** anatomy, lumbar spine, spinal canal, evolutionary changes, developmental stenosis, computed tomography

## Abstract

Spinal canal dimensions may vary according to ethnicity as reported values differ among studies in European and Chinese populations. Here, we studied the change in the cross-sectional area (CSA) of the osseous lumbar spinal canal measured in subjects from three ethnic groups born 70 years apart and established reference values for our local population. This retrospective study included a total of 1050 subjects born between 1930 and 1999 stratified by birth decade. All subjects underwent lumbar spine computed tomography (CT) as a standardized imaging procedure following trauma. Three independent observers measured the CSA of the osseous lumbar spinal canal at the L2 and L4 pedicle levels. Lumbar spine CSA was smaller at both L2 and L4 in subjects born in later generations (*p* < 0.001; *p* = 0.001). This difference reached significance for patients born three to five decades apart. This was also true within two of the three ethnic subgroups. Patient height was very weakly correlated with the CSA at both L2 and L4 (r = 0.109, *p* = 0.005; r = 0.116, *p* = 0.002). The interobserver reliability of the measurements was good. This study confirms the decrease of osseous lumbar spinal canal dimensions across decades in our local population.

## 1. Introduction

Age-related changes in facet joints, intervertebral discs and ligamentum flavum may lead to narrowing of the spinal canal causing acquired lumbar spinal stenosis (LSS), which increases the probability of lower limb pain and difficulty walking, thus limiting function and participation in daily activities [1,2,3,4]. This usually affects older adults and is the most frequent cause of elective lumbar spine surgery in this age cohort [5]. Lumbar spinal stenosis may be related to a pre-existing narrow osseous spinal canal due to abnormal development of the lumbar vertebrae with an additional degree of acquired narrowing secondary to degenerative spine disease [6,7]. It follows that the risk of developing symptomatic LSS could hypothetically be greater in patients with developmentally reduced lumbar cross-sectional area (CSA) of the osseous spinal canal [8]. While prevention of acquired spinal canal narrowing aims to limit degenerative changes during the life of an individual, determining the cause of developmental smaller spinal canal size remains challenging [9]. Several research studies focusing on the osseous spinal canal have measured lumbar bone CSA with computed tomography (CT) in specific populations, ultimately determining that generational changes have occurred [10,11,12]. One such study was based on North American and European populations using only two patient cohorts, one with subjects born in the 1970s and the other with subjects born in the 1940s [10,11]. Lumbar bone CSA was smaller in the younger group [10,11]. This unexpected finding was attributed to the increased maternal age and decrease in the prevalence of smokers in the younger patient cohort [13,14]. More recently, a study conducted in an ethnic Chinese population with a complete stratification of age groups found no change in the lumbar bone CSA between the age cohorts [12]. Considering the variability with these results is possibly related to heterogeneity within different ethnic populations, developing a reference range is required for our local population. Therefore, the goal of our study was to further investigate lumbar bone CSA in our geographic region and to more accurately define the evolution over time from patients born in the 1930s to those born in the 1990s in order to establish a reference range in our population. We hypothesized that lumbar bone CSA of the osseous spinal canal has been decreasing across birth decades over the past century. If a particular age group in our population shows significantly lower values of lumbar bone CSA, investigations and treatment of symptoms of LSS in this age cohort could theoretically be performed earlier. This would accelerate and improve the management of patients predisposed to LSS.

## 2. Materials and Methods

The local institutional ethics committee approved this single-center retrospective observational study, with a waiver of patient informed consent (IRB approval project-ID CER-VD 2020-02026, 29 October 2020). This study conforms to the STROBE guidelines for reporting observational studies.

### 2.1. Study Design and Patient Population

We retrieved a total of 1050 whole-body CT scans performed on 1050 patients following traumatic injury between January 2009 and December 2016. These examinations were randomly selected among a pool of a greater number of available CT scans in order to obtain cohorts of 150 patients per birth decade, creating seven cohorts from 1930–1939 to 1999–1999. As more male patients underwent CT examination for traumatic injury compared to female patients, each cohort included 110 males and 40 females, creating a matching male-to-female ratio of 2.75:1 for a total of 770 males and 280 females. Patients with lumbar spine fractures, previous spine surgery, scoliosis with a Cobb angle greater than 25 degrees, lumbosacral transitional abnormalities and motion or metal artifacts were not included in our sample.

The following information was collected from medical records when available: patient height (*n* = 676), weight (*n* = 676) and ethnicity (*n* = 1050). Ethnicity was categorized into three distinct groups: Swiss European (*n* = 648), non-Swiss European (*n* = 286) and non-European (*n* = 116).

### 2.2. CT Protocol

A 64- or a 256-detector row CT scanner (LightSpeed VCT or Revolution CT, respectively; GE Healthcare, Milwaukee, WI, USA) was used for all examinations with standardized acquisition parameters, which was optimized for whole-body trauma protocol. Patients were positioned in the supine position. Relevant acquisition parameters were as follows: tube potential, 120 kVp; tube current, ~300–400 mA with automatic exposure control enabled. Images were then reconstructed using the following parameters: section thickness/interval, 1.25/0.63 mm; pixel size, ~0.63–0.98 mm; sharp (Bone) convolution kernel.

### 2.3. Image Analysis

All acquired data were systematically analyzed on a picture archiving and communication system (PACS) workstation (Carestream Vue; Carestream Health, Rochester, NY, USA). Measurements were performed using a multiplanar reconstruction viewing mode in a plane perpendicular to the longitudinal axis of the spine and at the vertebral pedicle levels to limit the influence of degenerative changes on CSA measurements, which typically occur at the intervertebral disc and facet joint levels, as previously described (Figure 1). An attending musculoskeletal radiologist initially trained three independent observers (two orthopedic residents and a radiologist resident with two years and one year of experience in musculoskeletal imaging, respectively) by reading, in consensus, a sample set of ten cases not included in this study to define lumbar bone CSA and calibrate measurements. The three observers then independently measured the CSA of the osseous spinal canal by drawing the regions of interest (ROIs) at the L2 and L4 pedicle levels (Figure 1) [11,15]. These two vertebral levels were selected to provide a general overview of the lumbar spine while limiting the data collection efforts of the adjacent spinal levels. Once the ROI was traced using a free-hand tool, the surface area in square millimeters was automatically obtained and recorded in a spreadsheet. Each of the three observers measured 350 distinctive CT scans among which 50 cases were included from each decade with comparable male-to-female ratios in order to decrease measurement bias. In addition, each observer measured 30 random cases previously read by the two other observers in order to assess the interobserver reliability.

### 2.4. Statistical Analysis

All data were analyzed using SPSS statistics software (version 27.0; IBM SPSS Statistics for Windows, Armonk, NY, USA). Numeric (continuous) variables are presented as mean ± standard deviation (SD) and/or median, interquartile range (IQR), where appropriate. The normal distribution of the measurements was assessed using the Kolmogorov–Smirnov test and Q-Q plots. The lumbar bone CSA was compared between groups and subgroups using a dependent or independent two-sample Student’s t-test or a one-way analysis of variance (ANOVA) with Tukey’s post-hoc test, where appropriate. The correlation between lumbar bone CSA and patient height and weight was evaluated using Pearson’s correlation coefficient (r) and interpreted as follows: very weak, <0.20; weak, 0.20–0.39; moderate, 0.40–0.59; strong, 0.60–0.79; very strong, >0.79. The intraclass correlation coefficient (ICC; two-way random-effects model, single rater type, absolute agreement) was calculated to assess the interobserver reliability of the measurements and interpreted as follows: poor, <0.50; moderate, 0.50–0.74; good, 0.75–0.90; and excellent, >0.90 [16]. A significance level of *p* <0.05 was considered for all tests, after adjustment for multiple post-hoc comparisons (Dunn–Bonferroni method), where appropriate.

## 3. Results

### 3.1. Comparison between L2 and L4 Levels

Mean ± SD and median (IQR) lumbar bone CSA among all patients included in this study were 287.4 ± 44.4 and 282 (58) mm^2^ at the L2 level and 307.6 ± 63.9 and 300 (82) mm^2^ at the L4 level, respectively. A detailed breakdown of the osseous spinal canal CSA at L2 and L4 levels per birth decade is shown in Table 1 and Table 2, respectively. Analyzed at each birth decade, the mean CSA at L2 was significantly smaller than the mean CSA at L4 (all *p* < 0.001; Figure 2).

### 3.2. Birth Decade

Among all patients in this study, mean lumbar bone CSA decreased across decades at both L2 (*p* < 0.001) and L4 (*p* = 0.001) levels (Figure 2). With the exception of CSA at L4 in patients born in the 1950s, mean CSA at both L2 and L4 was always numerically smaller in subsequent birth decades. Moreover, when comparing across multiple birth decades, the differences in CSA reached pairwise statistical significance for patients born three to four decades apart for L2 and five decades apart for L4 (Table 3 and Table 4, respectively). The change in CSA from one decade to the next ranged from −1.2 mm^2^ (1950s–1960s) to −8.5 mm^2^ (1960s–1970s) for the L2 level and from +1.5 mm^2^ (1940s–1950s) to −9.1 mm^2^ (1930s–1940s) for the L4 level.

### 3.3. Sex

The male-to-female ratio was perfectly matched in all birth decade subgroups, as per our study design. Among all patients in this study, males had a significantly larger mean lumbar bone CSA than females at L4 (*p* = 0.004) but not at L2 (*p* = 0.358; Figure 3, Table A1, Table A2, Table A3 and Table A4 in Appendix A). With the exception of CSA in females born from the 1940s to the 1960s (*p* ≥ 0.128), mean lumbar bone CSA was significantly larger at L4 than at L2 (*p* ≤ 0.027). Among both males and females, a subgroup analysis showed that the CSA decreased across decades at both L2 (males, *p* = 0.001; females, *p* < 0.001) and L4 (males, *p* = 0.07; females, *p* = 0.007) levels (Figure 3, Table A1, Table A2, Table A3 and Table A4 in Appendix A).

### 3.4. Height and Weight

Mean ± SD height was 173.5 ± 8.0 cm. Mean ± SD weight was 77.2 ± 15.2 kg. Patient height was statistically, yet very weakly, significantly correlated with the CSA of the osseous spinal canal at both L2 (r = 0.109, *p* = 0.005) and L4 (r = 0.116, *p* = 0.002) levels (Figure 4). No correlation was found between patient weight and the lumbar bone CSA (r ≤ 0.061, *p* ≥ 0.111).

### 3.5. Ethnicity

Among all patients in this study, mean lumbar bone CSA was smaller in the later-born generations at both L2 (*p* < 0.001) and L4 (*p* < 0.001) levels in all three ethnic subgroups (Figure 5). In terms of numerical values, non-European subjects had smaller osseous spinal canals compared to both Swiss European and non-Swiss European subjects at both L2 and L4 levels (Figure 5, Table A5, Table A6, Table A7, Table A8, Table A9 and Table A10 in Appendix A). This difference reached pairwise statistical significance at L2 between non-European and Swiss European subjects born in the 1990s (*p* = 0.009). At the L4 level, non-European subjects had significantly smaller lumbar bone CSAs than Swiss European subjects when born in the 1960s (*p* = 0.011) and 1990s (*p* = 0.001) as well as non-Swiss European subjects born in the 1990s (*p* = 0.014) (Figure 5, Table A5, Table A6, Table A7, Table A8, Table A9 and Table A10 in Appendix A). Subgroup analysis failed to show any significant difference in the CSA between Swiss European and non-Swiss European subjects across decades at both L2 and L4 levels (*p* = 0.148).

### 3.6. Measurement Reliability

The interobserver reliability of the measurements was good with ICCs ranging from 0.832 (95% CI, 0.726–0.900) to 0.881 (95% CI, 0.803–0.930).

## 4. Discussion

The present study confirmed that subjects in later-born cohorts have smaller lumbar spinal canals at the pedicle level. Differences in lumbar bone CSA were accentuated as a function of the size of the generation gap and reached significance when this gap was three to five decades apart. The decrease in size of the CSA was observed at both levels investigated (L2 and L4) between almost every decade. Our study confirms the previously reported findings of smaller osseous lumbar spinal canals among younger patients and possibly eliminates any bias as it includes a larger sample size measured longitudinally [11]. When analyzing subgroups by sex and ethnicity, the CSA measured at L4 was larger than the CSA measured at L2 in most of the subgroups and this suggests that the CSA at the L4 level is greater than at the L2 level regardless of sex and ethnicity, as previously reported [10,12,15]. The decrease in CSA in subjects born in later-born cohorts compared to those subjects born a few decades earlier in the last century was found at both levels analyzed. This decreasing trend was observed in both men and women. The same trend was also observed in three different ethnic subgroups. A similar study using larger cohorts would probably confirm this trend statistically.

Similar to the results published by Schizas et al., Monier et al. and Tobin et al., we also demonstrated a decrease in size of the lumbar bone CSA in the younger generations [10,11,12,15]. Griffith et al., who analyzed a population of Chinese ethnicity, did not show any change in lumbar bone CSA between the different decades [12]. Tobin et al. subdivided their cohort into different ethnic groups and found a decrease in CSA in different population groups [10]. In the present study, we decided to allocate subjects in three separate subgroups according to their ethnicity. Although these subgroups differed in cohort size, there was a statistical trend for a decrease in CSA over time in all three subgroups (Swiss European, non-Swiss European and non-European) at both levels examined between subjects born in the 1930s–1940s and those born in the 1980s–1990s for example.

Spinal canal growth depends on neurocentral cartilage growth and can be influenced during the antenatal and early postnatal development period by factors such as protein intake [9]. However, protein intake has increased in the younger generations [17]; therefore, one would expect to observe a progressive enlargement of the osseous spinal canal in later-born generations. The explanation for the smaller osseous spinal canal in subjects from the later-born generations can only be speculated. One possible explanation is the change in lifestyle over time. For example, increased rates of maternal smoking could be a causative agent given that there is a correlation with lower birth weight and spinal canal size as well as neural tube defects [18,19]. The prevalence of female smoking has been on the rise for several decades since the beginning of the past century [20]. Another potential explanation is maternal age, which also influences spinal canal dimensions as older mothers have a higher risk of giving birth to infants with smaller spinal canals [21]. Maternal age has been increasing with time and may be an additional factor resulting in the decreased osseous spinal canal dimensions [22]. The lack of relationship between height and osseous spinal canal dimensions has already been described and is due to long bone growth continuing well into puberty, whereas spinal canal size reaches its final dimensions at a much earlier age [23,24].

Though variations in lumbar bone CSA in different ethnic groups have also been described, the observed differences between Swiss European and non-European subjects could be attributed to environmental factors rather than genetics [25,26]. Swiss European and non-Swiss European populations shared both a similar genetic extraction and similar environmental factors, and thus, both have larger spinal canal dimensions compared to the non-European subgroup, which is less genetically defined with unknown differences in environmental factors. Larger studies could help us better understand lumbar bone CSA differences based on geographic distribution and the extent of changes in CSA in some populations.

In theory, enlargement of the osseous spinal canal could occur due to aging and bone resorption, but this has not been reported. In addition, larger lumbar bone CSAs in the older generations were found in both males and females. Therefore, remodeling due to osteoporosis (more frequent in older female subjects), for example, is unlikely to explain this finding.

There are several limitations in this study. First, due to its retrospective nature, we lack information on nutritional and other antenatal parameters for each patient and were only able to speculate on the cause of our findings. Second, the ethnicity data could not be verified and may not accurately reflect any particular genetic difference between the three different ethnic populations presented. Longitudinal population-based cohort studies, including antenatal parameters, are necessary to confirm the hypothesis that current environmental and lifestyle changes are directly related to progressive narrowing of the osseous lumbar spinal canal in humans. Third, a measurement bias could theoretically be present, as data were derived from three different human observers. However, the reliability index of measurements was good for a relatively straightforward task. Finally, medical conditions that are known to affect bone remodeling of the spinal canal and dural ectasia, such as neurofibromatosis, Marfan syndrome, Ehler–Danlos syndrome, homocystinuria and ankylosing spondylitis, were not screened in our patient sample [27,28]. The impact of such conditions on spinal canal CSA remains limited as the pedicle vertebral level was selected for data measurements.

Given that patients requiring spinal decompression tend to have a narrower spinal canal at the pedicle level in addition to narrowing at the intervertebral disc level compared to controls, there is a possibility that more patients will present with symptomatic LSS requiring treatment aggravated by the aging of the general population [7]. Furthermore, this study supports the notion that patients may theoretically present with early-onset LSS leading to surgical decompression at a younger age.

## 5. Conclusions

Lumbar spinal canal CSA has decreased at both L2 and L4 pedicle levels in later-born generations over the past century and reached significance for patients born three to five decades apart. A reference range of lumbar bone CSA was established for our local population and showed inhomogeneity according to ethnicity. This may impact the management of patients developing symptoms of earlier-onset lumbar spinal stenosis as the later-born population ages.

## Figures and Tables

**Figure 1 diagnostics-13-00734-f001:**
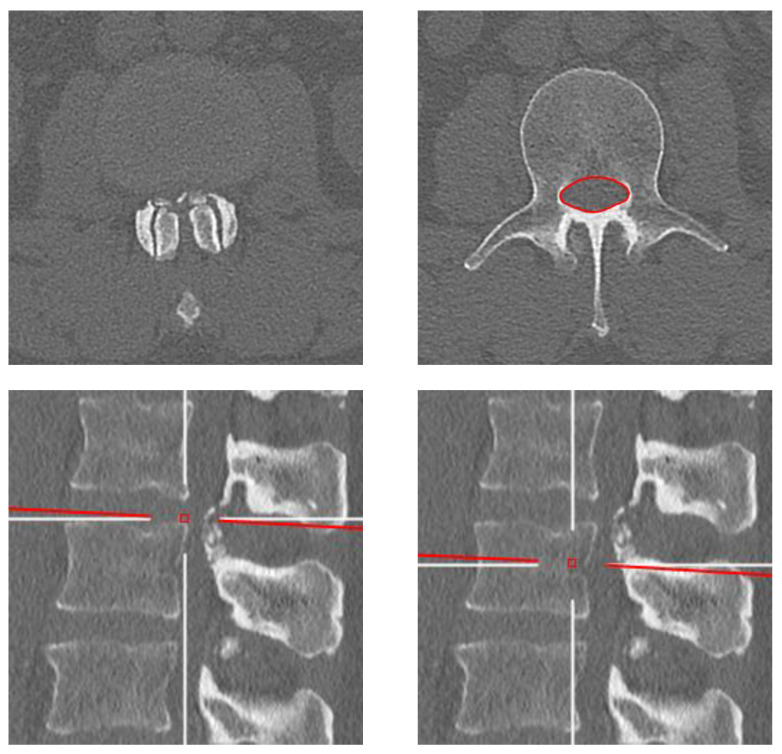
Representative axial (**top row**) and sagittal-reformatted (**bottom row**) computed tomography images of the lumbar spine at the L2 vertebral level from a male patient born in the 1970s. Note the change in lumbar bone cross-sectional area when measured at the vertebral pedicle level (**right column**) compared to the intervertebral disc level (**left column**) due to facet joint osteoarthritis and calcification of the ligamentum flavum in the latter.

**Figure 2 diagnostics-13-00734-f002:**
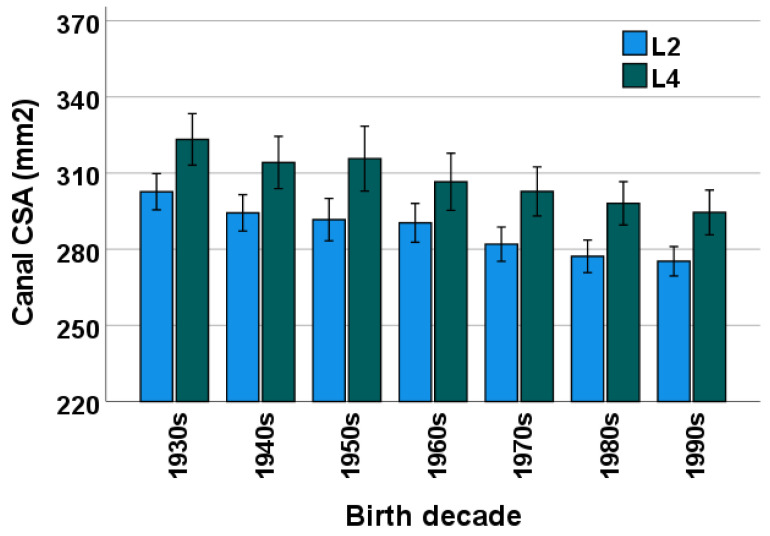
Mean ± SD (error bars) CSA of the osseous lumbar spinal canal across birth decades at the L2 vertebral level (blue) and the L4 vertebral level (green). SD = standard deviation, CSA = cross-sectional area.

**Figure 3 diagnostics-13-00734-f003:**
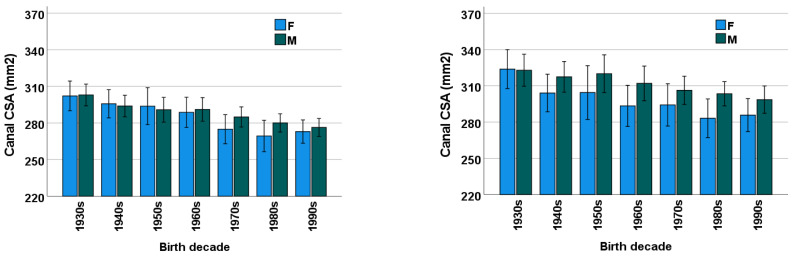
Mean ± SD (error bars) CSA of the osseous lumbar spinal canal across birth decades for females (F, blue) and males (M, green) at L2 (**left**) and L4 (**right**) vertebral levels. SD = standard deviation, CSA = cross-sectional area.

**Figure 4 diagnostics-13-00734-f004:**
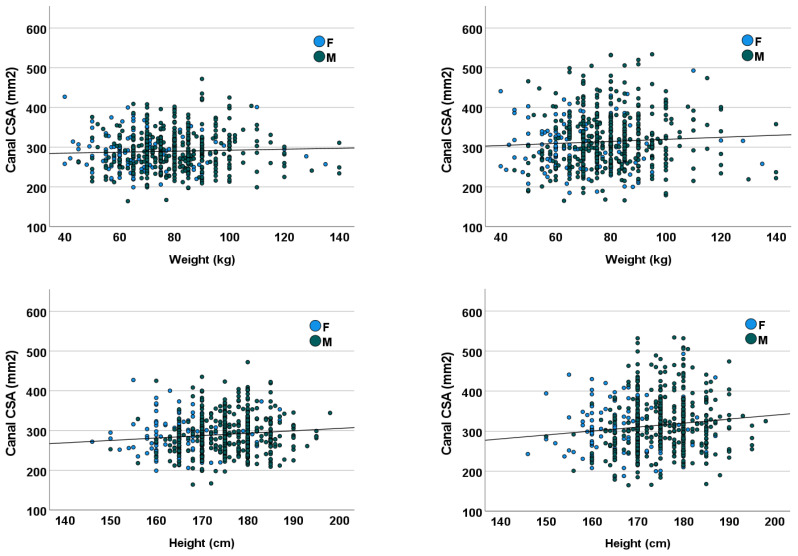
CSA of the osseous lumbar spinal canal as a function of patient height (**top row**) and weight (**bottom row**) for females (F, blue) and males (M, green) at the L2 (**left**) and L4 (**right**) vertebral level. CSA = cross-sectional area.

**Figure 5 diagnostics-13-00734-f005:**
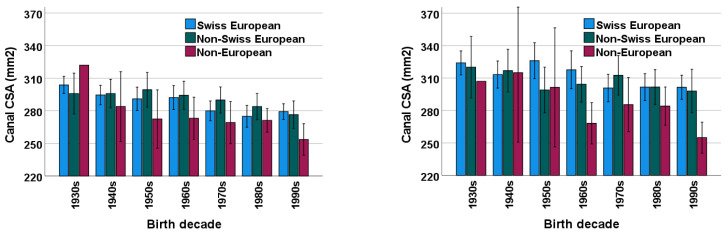
Mean ± SD (error bars) CSA of the osseous lumbar spinal canal across birth decades for Swiss European (blue), non-Swiss European (green) and non-European (purple) subjects at the L2 vertebral level (**left**) and L4 vertebral level (**right**). SD = standard deviation, CSA = cross-sectional area.

**Table 1 diagnostics-13-00734-t001:** Mean ± SD and median (IQR) CSA of the osseous lumbar spinal canal at the L2 vertebral level stratified according to birth decade.

Decade	Mean CSA ± SD (mm^2^)	Mean CSA − 2 SD (mm^2^)	Median CSA (IQR) (mm^2^)	Lower Quartile CSA (25th Percentile, Q1) (mm^2^)
1930s	302.6 ± 43.4	215.8	302 (54)	270
1940s	294.4 ± 45.7	203.0	294 (58)	266
1950s	291.7 ± 50.4	190.9	287 (68)	252
1960s	290.5 ± 46.2	198.1	284.5 (56)	260
1970s	282.0 ± 41.5	199.0	274 (54)	251
1980s	277.2 ± 39.9	197.4	274 (52)	248
1990s	275.3 ± 37.5	200.3	267 (52)	249

CSA: cross-sectional area, SD: standard deviation, IQR: interquartile range.

**Table 2 diagnostics-13-00734-t002:** Mean ± SD and median (IQR) CSA of the osseous lumbar spinal canal at the L4 vertebral level stratified according to birth decade.

Decade	Mean CSA ± SD (mm^2^)	Mean CSA − 2 SD (mm^2^)	Median CSA (IQR) (mm^2^)	Lower Quartile CSA (25th Percentile, Q1) (mm^2^)
1930s	323.3 ± 61.8	199.7	308 (96)	275.5
1940s	314.2 ± 65.7	182.8	306 (88)	269
1950s	315.7 ± 77.3	166.1	302 (96)	263.5
1960s	306.7 ± 67.8	171.1	297 (89)	255
1970s	302.8 ± 59.3	184.2	298 (80)	260.5
1980s	298.1 ± 53.0	192.1	297 (71)	261.5
1990s	294.5 ± 57.0	180.5	286 (84)	251.5

**Table 3 diagnostics-13-00734-t003:** Comparison of the lumbar bone CSA according to birth decade at the L2 vertebral level. *p*-values that reached statistical significance are highlighted.

Decade	1930s	1940s	1950s	1960s	1970s	1980s	1990s
1930s							
1940s	0.65						
1950s	0.34	0.99					
1960s	0.22	0.99	1.00				
1970s	**0.001**	0.17	0.49	0.64			
1980s	**<0.001**	**0.01**	0.07	0.12	0.97		
1990s	**<0.001**	**0.002**	**0.02**	**0.04**	0.83	1.00	

**Table 4 diagnostics-13-00734-t004:** Comparison of the lumbar bone CSA according to birth decade at the L4 vertebral level. *p*-values that reached statistical significance are highlighted.

Decade	1930s	1940s	1950s	1960s	1970s	1980s	1990s
1930s							
1940s	0.88						
1950s	0.95	1.00					
1960s	0.29	0.95	0.90				
1970s	0.09	0.70	0.59	0.99			
1980s	**0.01**	0.28	0.21	0.91	0.99		
1990s	**0.002**	0.08	0.06	0.63	0.91	0.99	

## Data Availability

The data presented in this study are available upon reasonable request from the corresponding author.

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
