# Peer review of "Evolution of the Cross-Sectional Area of the Osseous Lumbar Spinal Canal across Decades: A CT Study with Reference Ranges in a Swiss Population"

_diagnostics, 2023, doi:10.3390/diagnostics13040734_

Round 1

Reviewer 1 Report

Summary: The authors aimed to investigate lumbar bone CSA in a large cohort and to define the evolution of lumbar bone CSA from patients born in the 1930s to the 1990s in order to establish a reference range. In that sense, the study is very-well conducted, easy to understand present the results in a brief and concise way.

Title: OK

Abstract: OK

Introduction: Overall very good. I would suggest to formulate a hypothesis at the end of the introduction. What were the expectations of the authors? Were that expectations confirmed by their findings?

Materials and methods: OK.

Results: OK.

Discussion: OK.

Conclusion: Supported by the findings.

References: OK.

Figures: OK.

Tables: OK. 

Author Response

Summary: The authors aimed to investigate lumbar bone CSA in a large cohort and to define the evolution of lumbar bone CSA from patients born in the 1930s to the 1990s in order to establish a reference range. In that sense, the study is very-well conducted, easy to understand present the results in a brief and concise way.

Title: OK

Abstract: OK

Introduction: Overall very good. I would suggest to formulate a hypothesis at the end of the introduction. What were the expectations of the authors? Were that expectations confirmed by their findings?

Response: Thank you for your comments. We have added our hypothesis, now found on lines 60-62 in the introduction section. This sentence is as follows: “We hypothesized that lumbar bone CSA of the osseous spinal canal has been decreasing across birth decades over the past century.” Our results, which align very well with previously reported studies, supports our hypothesis. Being that our expectations were confirmed, we have opened our discussion section with the following sentence on lines 219-220:  “The present study confirmed that subjects in later-born cohorts have smaller lumbar spinal canals at the pedicle level.”

Materials and methods: OK.

Results: OK.

Discussion: OK.

Conclusion: Supported by the findings.

References: OK.

Figures: OK.

Tables: OK. 

Reviewer 2 Report

Thanks for a very concise review and well-presented data. The data seem real, and the experiment valid.

I wonder if you're too quick to dismiss the possibility that these are acquired changes with age. In the spinal canal, epidural pulsations are present, with likely billions by the time one reaches an old age. In dural ectatic conditions, epidural pulsations lead to acquired resorption of bone. (or some other process whereby the dura/venous/bone interplay leads to resorption). Could something similar / related be happening at low rates in humans?

Of course, if true, the more interesting question is whether we're going to be inundated with spinal stenosis cases as younger/smaller cohorts age. I would imagine the relationship between pedicle x-section area and facet/disc x-section area have a tenuous relationship in the CTs you've collected - that may be of interest.

Author Response

Thanks for a very concise review and well-presented data. The data seem real, and the experiment valid.

I wonder if you're too quick to dismiss the possibility that these are acquired changes with age. In the spinal canal, epidural pulsations are present, with likely billions by the time one reaches an old age. In dural ectatic conditions, epidural pulsations lead to acquired resorption of bone. (or some other process whereby the dura/venous/bone interplay leads to resorption). Could something similar / related be happening at low rates in humans?

Response: Thank you for your comment. We agree that certain diseases known to cause bone remodeling and subsequently dural ectasia, albeit rare, were not excluded from our patient population and could affect our results. The following sentences were added to the discussion section on lines 284-289: “Finally, medical conditions that are known to affect bone remodeling of the spinal canal and dural ectasia such as neurofibromatosis, Marfan syndrome, Ehler-Danlos syndrome, homocystinuria, and ankylosing spondylitis were not screened in our patient sample. The impact of such conditions on spinal canal CSA remains limited as the pedicle vertebral level was selected for data measurements.”

Of course, if true, the more interesting question is whether we're going to be inundated with spinal stenosis cases as younger/smaller cohorts age. I would imagine the relationship between pedicle x-section area and facet/disc x-section area have a tenuous relationship in the CTs you've collected - that may be of interest.

Response: Thank you for your comment. As we stated in the caption of Figure 1 on lines 117-122, we deliberately performed measurements at the pedicle x-section level in order to avoid erroneous measurements of the x-section at the facet-level where heterotopic calcifications at the facet joints for example are typically found. We additionally added a sentence in the methods section on lines 108-109 : “These two vertebral levels were selected to provide a general overview of the lumbar spine while limiting the data collection efforts of the adjacent spinal levels.”

Reviewer 3 Report

Aim is to investigate lumbar bone CSA in our geographic region and to more accurately define the evolution over time from patients born in the 1930s to those born in the 1990s  in order to establish a reference range in our population.

Introduction: it is not easy to understand your logic. Make three paragraphs; known, unknown, the purpose parts.

L66, Check the guideline, and cite it.  For example, STROBE.

L68, consecutive 1,050 patients?

L74,  describe the inclusion/ exclusion criteria clearly.

L101, Who is The three observers? Expert? Surgeon?

L133. Add the descriptive data of included patients.

Discussion: please write your strength points and clinical and research implication clearly.

Author Response

Aim is to investigate lumbar bone CSA in our geographic region and to more accurately define the evolution over time from patients born in the 1930s to those born in the 1990s  in order to establish a reference range in our population.

Introduction: it is not easy to understand your logic. Make three paragraphs; known, unknown, the purpose parts.

Response: Many thanks for your comments and suggestions. The known is elucidated upon in the introduction on lines 43-55. The unknown is located on lines 55-65 in the introduction. Finally, an additional sentence clearly stating the hypothesis of our study has been added on lines 60-61.

L66, Check the guideline, and cite it.  For example, STROBE.

Response: Thank you for your suggestion. The following sentence has been added to the materials and methods section on lines 69-70: “This study conforms to the STROBE guidelines for reporting observational studies.”

L68, consecutive 1,050 patients?

Response: In order to have evenly numbered cohorts with fixed male-to-female ratios for each birth decade, we needed to perform a random selection from an existing pool of available CT scans as more male patients underwent CT examination for traumatic injury compared to female patients.

L74,  describe the inclusion/ exclusion criteria clearly.

Response: Inclusion and exclusion criteria is stated clearly in the materials and methods section under section 2.1 study design and patient population on lines 71-85 : “We retrieved a total of 1,050 whole-body CT scans performed on 1,050 patients following traumatic injury between January 2009 and December 2016. These examinations were randomly selected among a pool of a greater number of available CT scans in order to obtain cohorts of 150 patients per birth decade, creating seven cohorts from 1930-39 to 1999-99. As more male patients underwent CT examination for traumatic injury com-pared to female patients, each cohort included 110 males and 40 females creating a matching male-to-female ratio of 2.75:1 for a total of 770 males and 280 females. Patients with lumbar spine fractures, previous spine surgery, scoliosis with a Cobb angle greater than 25 degrees, lumbosacral transitional abnormalities and motion or metal artifacts were not included in our sample. The following information was collected from medical records when available: patient height (n=676), weight (n=676), and ethnicity (n=1,050). Ethnicity was categorized into three distinct groups: Swiss European (n=648), non-Swiss European (n=286) and non-European (n=116).”

L101, Who is The three observers? Expert? Surgeon?

Response: To better clarify the experience level of each of the the three observers, we have modified a sentence in section 2.3 image analysis on lines 102-106: “An attending musculoskeletal radiologist initially trained three independent observers (two orthopedic residents and a radiologist resident with two years and one year of experience in musculoskeletal imaging, respectively) by reading, in consensus, a sample set of ten cases not included in this study to define lumbar bone CSA and calibrate measurements.”

L133. Add the descriptive data of included patients.

Response: Many thanks for your suggestion. We have included a detailed description of the patients included in this study in the results section starting on line 139 of the submitted manuscript. Moreover, the following sentence on lines 78-81 in the methods section, which states our exclusion criteria, describes further our patient population, “Patients with lumbar spine fractures, previous spine surgery, scoliosis with a Cobb angle greater than 25 degrees, lumbosacral transitional abnormalities and motion or metal artifacts were not included in our sample.”

Discussion: please write your strength points and clinical and research implication clearly.

Response: Our study confirms the findings of smaller osseous lumbar spinal canals among younger patients previously reported and possibly eliminates any bias as it includes a larger sample size measured longitudinally as stated on lines 219-225 in the discussion section. A reference range of L2 and L4 cross-sectional area for our local population has been obtained. Finally, we allude to the possible clinical impact of our study at the end of the discussion section and additionally, have included another sentence on lines 293-295: “Furthermore, this study supports the notion that patients may theoretically present with early-onset LSS leading to surgical decompression at a younger age.”

Reviewer 4 Report

This is an exciting study. Knowing the differences in the spine canal measures between different populations and age groups could help speculate future trends in spine surgery.

I have some concerns for the authors:

1)     Why L2 and L4 were chosen for CSA analysis?

2)     please state in the discussion section the implication of the study results from a therapeutic point of view. Are there any possible changes in the future in the incidence of degenerative spine conditions linked to the narrowing of the spinal canal trend?

3)     Please state in the limitation section that one limitation is the fact that CT scans were taken in the supine position. Is it possible that spine canal diameter changes in standing position?

Author Response

This is an exciting study. Knowing the differences in the spine canal measures between different populations and age groups could help speculate future trends in spine surgery.

Response: Many thanks for your comments. We speculated one ramification of our study in clinical practice in the conclusion as stated on lines 299-302: “A reference range of lumbar bone CSA was established for our local population and showed inhomogeneity according to ethnicity. This may impact the management of patients developing symptoms of earlier-onset lumbar spinal stenosis as the later-born population ages.”

I have some concerns for the authors:

1)    Why L2 and L4 were chosen for CSA analysis?

Response: Thank you for your feedback and comment. We decided to perform measurements at two non-adjacent levels to cover, in a general fashion, the lumbar spine, while optimizing human resources for data collection. Ideally, measurements at all five lumbar levels would provide us with the most amount of data, but unfortunately would be too labor-intensive in 1’050 subjects. The L5 level was not chosen as it presents higher variability in cross sectional area as previously described by Schizas. et al. (in our references). The L4 level represents a level in the lower lumbar spine and thus was used in our analysis. In order to cover another region of the lumbar spine while not using L3, the adjacent level, we selected L2. Therefore, we decided to perform our measurement at L2 and L4 levels. The following sentence has been added to the methods section on lines 108-109: “These two vertebral levels were selected to provide a general overview of the lumbar spine while limiting the data collection efforts of the adjacent spinal levels.”

2)   Please state in the discussion section the implication of the study results from a therapeutic point of view. Are there any possible changes in the future in the incidence of degenerative spine conditions linked to the narrowing of the spinal canal trend?

Response: The study shows that later-born cohorts (younger patients at data collection) have smaller spinal canals. Although these patients may be asymptomatic at the time we performed measurements, they may be at risk of developing early-onset symptomatic lumbar spinal stenosis. An increase incidence of spinal decompression surgery and/or a shift in the median age for such a procedure remains to be seen and could ultimately confirm these findings. Additionally, there are some differences across ethnicities, which may additionally be considered as a risk factor for developing symptoms. The following sentence has been added the discussion section of the manuscript on lines 293-295: “Furthermore, this study supports the notion that patients may theoretically present with early-onset LSS leading to surgical decompression at a younger age.”

3)     Please state in the limitation section that one limitation is the fact that CT scans were taken in the supine position. Is it possible that spine canal diameter changes in standing position?

Response: Thank you for your suggestion. We believe that lumbar cross sectional area would remain unchanged regardless of patient position as we deliberately performed measurements at the pedicle level. These measurements would certainly change if measurements were taken at the facet or disco-vertebral level. As we stated in the caption of Figure 1 on lines 117-122, we deliberately performed measurements at the pedicle level.

Round 2

Reviewer 3 Report

Thank you for revision